# Influence of the Carbon Fiber Length Distribution in Polymer Matrix Composites for Large Format Additive Manufacturing via Fused Granular Fabrication

**DOI:** 10.3390/polym16010060

**Published:** 2023-12-23

**Authors:** Pedro Burgos Pintos, Daniel Moreno Sánchez, Francisco J. Delgado, Alberto Sanz de León, Sergio I. Molina

**Affiliations:** Departamento de Ciencia de los Materiales e Ingeniería Metalúrgica y Química Inorgánica, IMEYMAT, Facultad de Ciencias, Universidad de Cádiz, Campus Río San Pedro, 11510 Puerto Real, Cádiz, Spain; pedro.burgos@uca.es (P.B.P.); danielmoreno.sanchez@uca.es (D.M.S.); fjdelgado@uca.es (F.J.D.); sergio.molina@uca.es (S.I.M.)

**Keywords:** large format additive manufacturing, fused granular fabrication, polymer matrix composites, carbon fiber, mechanical properties, functional properties

## Abstract

Many studies assess the suitability of fiber-reinforced polymer composites in additive manufacturing. However, the influence of the fiber length distribution on the mechanical and functional properties of printed parts using these technologies has not been addressed so far. Hence, in this work we compare different composites based on Acrylonitrile Styrene Acrylate (ASA) and carbon fiber (CF) suitable for large format additive manufacturing (LFAM) technologies based on fused granular fabrication (FGF). We study in detail the influence of the CF size on the processing and final properties of these materials. Better reinforcements were achieved with longer CF, reaching Young’s modulus and tensile strength values of 7500 MPa and 75 MPa, respectively, for printed specimens. However, the longer CF also worsened the interlayer adhesion of ASA to a greater extent. The composites also exhibited electrical properties characteristic of electrostatic dissipative (ESD) materials (10^5^–10^10^ Ω/sq) and low coefficients of thermal expansion below 15 µm/m·°C. These properties are governed by the CF length distribution, so this variable may be used to tune these values. These composites are promising candidates for the design of elements with enhanced mechanical and functional properties for ESD protection elements or molds, so the products can be manufactured on demand.

## 1. Introduction

Additive manufacturing (AM) has revolutionized industrial transformation processes by being able to generate custom-made objects, greatly reducing the overall product development and manufacturing time and leading to a quicker transfer to market, allowing on-demand manufacturing. AM approaches enable flexible production and mass customization so designs can be changed quickly, allowing the reduction of the final cost of the products [1]. The technologies known as large format additive manufacturing (LFAM) are of special interest in industrial sectors, such automotive, naval, aerospace or construction, where the fabrication of large parts (typically bigger than 1 m^3^) with higher deposition rates is required [2,3,4]. There are currently LFAM systems that can reach printing rates of more than 45 kg/h and print structures of volumes larger than 27 m^3^ [5]. Among the LFAM technologies, the most popular are the extrusion-based ones, in particular, fused filament fabrication (FFF) and fused granular fabrication (FGF). In FFF, a filament is extruded through a previously heated nozzle that allows its melt deposition. The FGF technology is similar to FFF but uses pellets as feedstock material. In these systems, an extrusion screw rotates inside the previously heated printing head and melts the pellets, turning them into a homogeneous mass, which is pushed through a nozzle without the need to previously have a filament. In LFAM systems, FGF presents a series of advantages compared to FFF: (1) it allows us to reduce production times, (2) raw materials are less expensive as there is no need to manufacture a filament, and (3) it offers greater versatility when using fiber-reinforced composites [6,7].

Fiber-reinforced composites allow us to obtain materials with superior performance to that of the original polymer, often with functional properties as well. In particular, carbon fiber (CF) is a very popular filler because it can increase the stiffness and strength of the polymer matrix without implying a significant increase in weight. Moreover, CF can also tune the thermal and electrical properties of the polymer in which it is embedded [8,9]. Longer CF always provide better reinforcement in both classical manufacturing (e.g., injection molding, compression molding) and AM, having more than one order of magnitude of difference in tensile strength when continuous, aligned CF is used [10]. However, longer CF also limits the processability of the composite. The presence of CF decreases the melt flow rate (MFR) of the material, which can lead to clogging and defective printing. Tekinalp et al. [11] reported these clogging issues for ABS composites loaded with 40 wt.% CF, indicating that composites with lower contents could be printed without major problems. Nonetheless, if the conditions are not optimized, clogging and other printing defects can be observed for 5 wt.% CF composites [2]. When the CF is short enough, the decrease in the MFR is minimal, allowing a correct processing, but this also leads to smaller improvements in the mechanical and functional properties, especially when they are randomly distributed. Therefore, a compromise between size, processability, and reinforcement must be reached [12]. In addition, the effect of the CF on the anisotropy of the mechanical and functional properties must also be considered when working with FFF and FGF technologies. CF tends to align in the printing direction [13], which increases the already existing anisotropy in mechanical properties that are characteristic of these technologies due to their poor interlayer adhesion [14,15,16]. Moreover, different reports show that the coefficient of the thermal expansion (CTE) of the material strongly depends on the CF direction for FFF and FGF printed parts [17,18]. This is a key property in the design of tools, molds, and dies for low service temperatures (i.e., below 100 °C), which have become one of the most promising applications of LFAM [19].

Among other parameters, it is well-known that the CF length distribution affects the mechanical properties of the composites [20,21]. It must be considered that the shear forces of the compounding processes break and therefore decrease the size of the CF, worsening the reinforcement. This has been assessed by Lee et al. [22] in ABS-based composites, where they also studied the influence of the compatibility of the A, B, and S blocks with the CF. The effect of the fiber length distribution in the mechanical properties has widely been studied for different matrixes in classical manufacturing. For instance, Zhang et al. [23] observed that longer CF increased not only the stiffness and strength of epoxy-based composites but also increased their wear resistance. The influence of fiber size on the functional properties has also been assessed. Yi et al. [24] discussed the influence of the fiber length distribution in their dispersibility and how that affected the thermal conductivity of the composite. Different authors also proved that the CF length distribution has a strong influence on the percolation limit at which the composite goes from being an insulator to having (semi)conductive properties. This limit is reached at values ranging from 2 to 14 vol.%, so lower CF concentrations are required when longer CF are used. This was observed for different polymer matrixes, including ABS, PMMA, and PP [25,26,27].

However, to the best of our knowledge, the influence of the CF length distribution on the mechanical and functional properties of composites manufactured by FGF have not been systematically studied yet. Hence, in this work we evaluate the influence of the CF size distribution in the mechanical and functional properties of ASA-based composites. This work proves that the CF length distribution has a major influence on the mechanical properties, electrical resistivity, and CTE of printed parts by FGF, proving that these properties can be tuned by varying the CF length distribution for the fixed CF content.

## 2. Materials and Methods

### 2.1. Materials

Acrylonitrile Styrene Acrylate (ASA) and two carbon fiber (CF)-reinforced ASA composites (labeled in this work as ASA-CF1 and ASA-CF2) were supplied in the form of pellets by Matersia (Cádiz, Spain). According to the supplier, both composites have a CF content of 23 wt.%, but ASA-CF1 has a shorter average fiber length than ASA-CF2. The fiber length distribution was quantified as indicated below in the characterization of the feedstock material section. All the materials were suitable for injection molding and additive manufacturing via melt extrusion.

### 2.2. Manufacturing of Standard Specimens

ASA, ASA-CF1, and ASA-CF2 were dried at 80 °C for 4 h prior to their use as feedstock in a large format fused granular fabrication (FGF) Bárcenas CNC Discovery 3D printer (Ciudad Real, Spain), equipped with a 2 mm diameter nozzle. Horizontal and vertical panels of 200 mm × 200 mm and 2 mm thickness were printed in the XY and XZ plane according to ISO 17295:2023 [28]. The horizonal panels were printed at a speed of 50 mm/s while the vertical panels were printed at 25 mm/s. A layer height of 1 mm and an extrusion temperature profile of 235/240/245 °C was used. The FGF extruder has 3 heating zones, with the lowest the one corresponding to the heater closest to the printing nozzle. The platform temperature was set to 100 °C in all cases, while the rest of the printing parameters were left as default. The 1BA tensile testing specimens and 65 mm diameter flat discs for electric conductivity measurements were cut from these panels using a computer numerical control (CNC) LEKN(C1) 3020 CNC Router Machine Kit, equipped with a 2 mm diameter flat-end mill with two cutting edges working at a speed of 5000 rpm and a feed rate of 300 mm/min. Similar panels with a 5 mm thickness were also printed and the CNC machine was used to obtain parallelepipeds of 20 mm × 5 mm × 5 mm for thermomechanical analysis. An 80 × 80 × 60 mm^3^ vase with different surface textures was also printed as a prototype. Tensile testing specimens were also manufactured via injection molding in a Babyplast 10/12P machine (Rambaldi+CO I.T. Srl, Molteno, Italy). The temperature profile of the injector was 230–240–235 °C in the plasticization, chamber, and nozzle areas, respectively.

### 2.3. Characterization of the Feedstock Material

To quantify the size distribution of the CF of the different composites used as feedstock, 30 mg of pellets were dissolved in 1 mL chloroform (purchased from Scharlau, Barcelona, Spain) and were centrifuged at 6000 rpm for 5 min in a miniG centrifuge (IKA, Staufen, Germany). The supernatant solution was removed and the sedimented CF was redispersed in fresh chloroform. This process was repeated three times to ensure the complete removal of ASA. Then, a droplet of the CF dispersion was deposited on a glass slide and the CF was observed under a Nikon MA100 inverted light microscope (Nikon, Tokyo, Japan) after evaporation of the solvent. The quantification of the CF size was carried out using the ImageJ 1.8.0 analysis software. In both cases, at least 250 different fibers were measured. The thermal stability and fiber content of the materials was examined by thermogravimetric analysis (TGA) in a Q50 (TA Instruments, New Castle, DE, USA). Following a typical procedure, a temperature sweep of up to 600 °C was performed using a constant rate of 10 °C/min. All the TGA experiments were carried out under a constant nitrogen flow. The melt flow rate (MFR) values of ASA, ASA+CF1, and ASA+CF2 were obtained in a LR-A001-A (Lonroy, Dongguan, China). The measurements were carried out by applying a load of 5 kg for 15 s at 245 °C. At least 3 independent measurements were performed to ensure the reproducibility of the results.

### 2.4. Characterization of the 3D-Printed Composites

The mechanical behavior of the materials was assessed using tensile testing experiments, performed in an AGS-X universal mechanical testing machine (Shimadzu, Kyoto, Japan) at 1 mm/min, in agreement with ASTM D638 standard [29]. Five samples of each material were tested to ensure the reproducibility of the results. The fracture surface of the already tested specimens were examined via scanning electron microscopy (SEM) in a FEI Nova NanoSEM 450 (Hillsboro, OR, USA), working at 1.50 kV. The samples for SEM were previously coated in a Balzers SCD 004 Sputter Coater (Oerlikon Balzers, Buffalo, NY, USA) with a few nanometers layer of gold. The electrical resistivity of ASA, ASA+CF1, and ASA+CF2 was measured in a Keithley 6517B electrometer (Keithley, Cleveland, OH, USA), using a voltage of 500 V, following the ASTM D257 standard [30]. Three surface and volume resistivity independent measurements were taken for each material. A PT1000 dilatometer (Linseis, Selb, Germany) was used to determine the coefficient of thermal expansion (CTE) through thermomechanical analysis (TMA). For this, at least 3 specimens of 5 × 5 × 20 mm^3^ for each material were subjected to a heating rate of 5 °C/min from 20 °C to 80 °C and their variation in length was measured.

## 3. Results and Discussion

### 3.1. Analysis of the Feedstock Material and Their Suitability for FGF-LFAM

The CF distribution of the composite pellets was observed under an inverted light microscope. Figure 1 shows representative optical micrographs of the CF from ASA+CF1 (Figure 1a) and ASA+CF2 (Figure 1b), where it is clearly observed that they possess the same diameter size (5 ± 1 µm) but different length distributions. Figure 1c shows the distribution size of the CF in both cases, proving that the CF in ASA+CF1 has an average size (42 ± 41 µm) that is significantly lower than those in ASA+CF2 (96 ± 57 µm). The fiber size distribution was fitted to a Lorentz function curve since this approach takes more account of the tails in a population distribution. Although both CF length distributions are broad, they present ranges like those previously studied by other authors [23,26,31].

Complementary TGA assays were carried out to check the amount of CF in the composites. Figure 2a shows that, after the removal of the ASA polymer at ca. 400 °C, a residue of ca. 23 wt.% is obtained in both composites, in good agreement with the information provided by the manufacturer. Hence, this proves that the differences observed hereafter are not caused by an effect of a different amount of CF. As expected, an increase in the thermal degradation of the composites when compared to pure ASA is observed. The derivative thermogravimetric (DTG) curves presented in Figure 2b show that the maximum degradation rate is shifted from 363 °C for ASA to 418 °C for ASA+CF1 and 407 °C for ASA+CF2 due to the presence of CF.

The MFR of ASA, ASA+CF1, and ASA+CF2 was also measured at 245 °C. The results obtained are presented in Table 1, which shows a significant decrease in the MFR of ASA when the CF is introduced. The presence of longer CF sizes in ASA+CF2 causes a much higher decrease in the MFR than in ASA+CF1, even though the amount of CF is the same. These values are above 10 g/10 min, the threshold value for processing materials by FFF or FGF, proving that both composites are, in principle, valid for these additive manufacturing technologies and can be successfully printed at 245 °C [32,33,34].

Once the different materials have been characterized and proven that they are all suitable for FGF, various objects were printed. For each material (ASA, ASA+CF1, and ASA+CF2), panels of 2–5 mm thickness were horizontally and vertically printed (XY and XZ specimens, respectively). Then, standard specimens for tensile testing, electric conductivity and TMA were cut out of these panels using a CNC machine (see the Materials and Methods section for more details). This was carried out to effectively test the mechanical or functional properties of the materials in the XY and XZ planes avoiding defects or discontinuities caused by the shell or infill during the FGF process, as was already carried out in previous studies [33]. ASA and ASA+CF1 presented good rheological behavior when printing, which was expected, given their MFR values. The standard specimens were easily cut using CNC, without the generation of small plastic chips that stick to the cutter. An illustrative example of 200 mm × 200 mm × 2 mm panels printed in the XY and XZ plane using ASA+CF1 are presented in Figure 3a,b. Moreover, Figure 3c shows an exhibition vase to illustrate the surface finish of the material with different textures. ASA+CF2 horizontal panels were also printed normally, and the standard specimens were obtained without problems. However, it was necessary to reduce the printing speed by 80% in order to properly print the vertical panels. Even with this modification, the adhesion between the layers was rather poor, presenting fracture and delamination of the material when cutting the specimens by CNC. This was likely due to the low MFR of ASA+CF2 at 245 °C, close to 10 g/10 min, as we had previously observed for ASA composites [2]. In any case, it was also possible to obtain the specimens for further testing, but the manufacturing was carried out in a more delicate way.

### 3.2. Influence of the CF Length Distribution in the Mechanical Properties of the Composites Manufactured by FGF-LFAM

The mechanical properties of the materials were obtained via tensile testing. For comparative purposes, the mechanical properties of these materials prepared using injection molding were also obtained. A summary with representative curves of specimens prepared via injection molding and FGF for ASA, ASA+CF1, and ASA+CF2 is presented in Figure 4. It must be noted that for FGF, the mechanical properties of the materials are presented for specimens printed horizontally (XY samples, Figure 4b) and vertically (XZ samples, Figure 4c). The first depicts the actual mechanical properties of the material obtained by FGF, while the latter accounts for the interlayer strength of the material during the printing process. A summary of the mechanical properties (Young’s modulus, tensile strength, and elongation at break) dissected from the stress–strain curves of ASA, ASA+CF1, and ASA+CF2 is presented in Table 2, Table 3 and Table 4. It should be noted that the Young’s modulus was measured as the slope at 0.05–0.25% strain for all the samples except for ASA+CF2 FGF_XZ, which was measured up to its elongation at break value, since these specimens broke at slightly lower values.

The curves for injected samples are quite similar to those of the XY-printed samples for ASA, ASA+CF1, and ASA+CF2. The stiffness and strength present similar values in all these cases, except for the tensile strength of ASA+CF2, which is ca. 20 MPa higher for injected specimens. Furthermore, the Young’s modulus of ASA+CF2 is higher for the XY-printed samples than for the injected ones. This may be related to the greater alignment of the CF caused by the FGF manufacturing process, which may contribute to a greater stiffness [35]. When ASA+CF1 and ASA+CF2 are compared, it is clear that ASA+CF2 has better mechanical properties for both injected and XY-printed specimens. The longer CF of ASA+CF2 can contribute to a larger extent to the fracture mechanism, withstanding higher loads when ASA fails. For both injected and XY-printed specimens, there are significantly higher Young’s modulus and strength values without a major decrease in the elongation at break when compared to ASA+CF1, indicating that, in principle and in agreement with the literature, longer CFs are desirable for higher reinforcements for a fixed amount of CF [27].

The elongation at break is significantly lower for XY-printed samples in all cases, however. This is particularly clear in the case of pure ASA, which shows a much lower plastic deformation. The decrease in the elongation at break also happens for both ASA+CF composites, but to a lesser extent, since they do not practically present plastic deformation in either injected or XY-printed samples leading to an embrittlement of the material. These differences can be attributed to the FGF printing process itself, which may favor the presence of small gaps interfaces between printing roads and layers that cause the initiation of fracture cracks.

When comparing the mechanical properties of the XZ samples, it can be observed that they are always significantly smaller than those of XY samples for all the materials studied. This is something characteristic and inherent of the extrusion-based AM processes, due to the weak interlayer adhesion. There are different approaches to overcome this issue, even though this is still a current challenge in the development of extrusion-based AM technologies [36,37], but the presence of CF is not reported to enhance the adhesion between layers. The failure observed in these samples is likely caused by the insufficient bonding between the CF and the ASA matrix. Therefore, the CF may act as potential surfaces for the initiation of cracks, even if a proper sizing has been applied to enhance their compatibility with the polymer matrix. This caused the material to fail at strengths even below than pure ASA, indicating that this failure took place at the interface between the ASA matrix and the CF. This has also been observed previously by us [38] and others [39,40] in short and continuous fiber reinforced composites printed by FFF and FGF. The anisotropy of the mechanical properties is bigger for ASA+CF2, which possess longer CF. They are likely to contribute to a larger extent to enhance the mechanical properties in the XY plane but limit the adhesion of two consecutive layers of ASA in the XZ plane during the printing process. This leads to a difference in maximum tensile strength from 74.8 to 1.2 MPa when the printed XY and XZ specimens are compared. This difference is diminished for ASA+CF1, which decreases from 65.4 to 6.5 MPa. Therefore, it is essential to identify the working loads to which the structural elements manufactured by FGF will be subjected to avoid material failures in service in addition to taking into account the influence of the CF content and length distribution.

The SEM micrographs of the fracture surface of the tensile specimens manufactured via FGF support the mechanical properties results observed. Figure 5a shows that the CF contribute to the fracture surface or the XY specimens (pull-out effect). This behavior is the characteristic of fiber-reinforced polymer matrix composites and has been previously observed in composites obtained via classical manufacturing and AM [35,41]. It can be observed that the length of the fibers coming out of the ASA surface is higher in the case of ASA+CF2 when compared to ASA+CF1, confirming that they can bear higher loads. This leads to the increased stiffness and tensile stress of ASA+CF2, as observed in Figure 4 and in agreement with other previous works [27,38,42]. The fracture surface of the XZ specimens is presented in Figure 5b. A much flatter surface is observed in this case, indicating that this is an elastic fracture, which is characteristic of brittle materials. The fracture of these specimens originated because two consecutive printed layers were pulled apart. The CF observed does not come from the polymer matrix surface, as in Figure 5a. Instead, they are displaced perpendicularly, not contributing to the fracture mechanism. On the contrary, they interfere with the adhesion between layers of ASA, decreasing the stiffness, the maximum strength, and the elongation at break. This proves that the CF does not contribute to the enhancement of the mechanical properties on the XZ plane.

### 3.3. Influence of the CF Length Distribution in the Functional Properties of the Composites Manufactured via FGF-LFAM

The electrical conductivity of the materials was also investigated. Surface and volume resistivity values of ASA, ASA+CF1, and ASA+CF2 are presented in Figure 6a,b. They show a drastic decrease in several orders of magnitude from pure ASA to any of the ASA+CF materials. Both composites present typical values of semiconductors, indicating that the CFs reached the percolation threshold, where a polymer matrix composite is no longer non-conductive. This agrees with previous reports, considering that 23 wt.% CF corresponds to approximately 16.2 vol.% and that different authors observed that the percolation limit was always reached for values lower than 14 vol.% when using CFs of similar size to those reported in our study [26,27].

Moreover, ASA+CF1 possesses electrical resistance values of several orders of magnitude lower than ASA+CF2, indicating that the fiber size plays a key role in the design of materials when tuning the electrical properties. This may seem counterintuitive, since longer fibers may reach the percolation threshold with lower contents [25,26], but we hypothesize that the electrical properties of the composites are also influenced by the FGF printing process. As depicted in Figure 6c, the CF have a preferential disposition within the composite, tending to be aligned in the printing direction. However, it is also observed that the shorter CF may have a higher tendency to deviate from this trend, being partially misaligned. These shorter fibers, displaced in a more random way, may act as bridges between longer CF, facilitating the transport of electrons, and therefore decreasing the electrical resistivity of the material for a fixed amount of CF. This would explain the lower resistivity values for ASA+CF1. In any case, both composites lie in the range of ESD materials, highlighted in light yellow in Figure 6a,b. This indicates that both composites can be used in the design of elements of protection for electric devices or staff, or to contain flammable liquids reducing the static electricity in industrial environments [43,44].

Finally, the thermomechanical properties of the FGF-printed materials were measured using TMA. It is well known that the presence of CF drastically decreases the CTE when used as a filler in a polymer matrix. This effect is presented in Figure 7, where both ASA+CF1 and ASA+CF2 present much lower dilatation values than ASA.

The CTE values measured at 70 °C for each of these materials are presented in Table 5. The CTE of ASA+CF2 is lower than that of ASA+CF1, likely due to the longer size of the CF, but these differences are rather small. Hence, in this case the fiber length distribution is not as critical as the presence of CF itself, which vastly limits the mobility of the ASA polymer chains when the temperature is increased. Moreover, no differences in dilatometry between XY and XZ specimens were observed. These results were unexpected, since we recently observed that ASA composites manufactured via FFF and reinforced with short fibers show anisotropy in the CTE [38], and other authors reported similar results [13,18,19]. However, the length of the specimens studied is only 20 mm and the layer height (1 mm) is significantly higher than in standard FFF or FGF, where values of 0.1–0.2 mm are typically used. This is a relatively short dimension that does not allow observing deviations associated with poor interlayer adhesion, so it may be that the CTE in the XZ and XY planes are similar in LFAM, as long as the manufactured objects are small. More importantly, the CTE values presented in Table 5 are below 15 and 8 µm/m·°C for ASA+CF1 and ASA+CF2, respectively. These values are in the range of metals that are used to make molds, such as aluminum or steel, proving that these materials could be suitable in these applications as an alternative, where a custom-made mold can be 3D-printed using a composite, consuming less energy and time, and saving costs [45].

## 4. Conclusions

In this work, we prove the influence of the CF length distribution in composites for FGF in LFAM. For this purpose, two composites (ASA+CF1 and ASA+CF2) with the same fiber content (23 wt.%) but different size distributions were tested. All the materials were successfully printed, although ASA+CF2 presented some problems due to the longer CF size. The mechanical properties, electrical resistivity, and CTE of these composites were studied and compared with the unmodified ASA. For XY-printed specimens, and an increase in the maximum tensile strength above 20 and 40 MPa was observed for ASA+CF1 and ASA+CF2, respectively. ASA+CF2 also presented a greater increase in stiffness and strength than ASA+CF1, as expected, because longer CFs act as better reinforcements. However, when the mechanical properties were evaluated in the XZ plane, ASA+CF2 had worse mechanical properties than ASA+CF1, because the longer CF worsened the adhesion between layers of ASA during the FGF process. Both composites presented an electrical resistivity characteristic of semiconductors, although differences of several orders of magnitude were observed between the two composites, due to the different fiber size. Finally, the CTE decreased significantly in both composites (below 15 and 8 µm/m·°C for ASA+CF1 and ASA+CF2, respectively), proving that the very presence of CF affects this property to a greater extent than their length distribution. The CTE values observed prove that these materials are of potential interest for the development of ESD materials or molds for low service temperatures, currently attained with metals.

Overall, this study demonstrates that fiber size, hitherto not studied in detail for FGF or FFF systems, is a critical parameter in the development of composites for LFAM. In the future, we plan to work on improving the adhesion between layers, since the composites herein presented are industrially interesting only when there are no external tensile loads on the XZ plane.

## Figures and Tables

**Figure 1 polymers-16-00060-f001:**
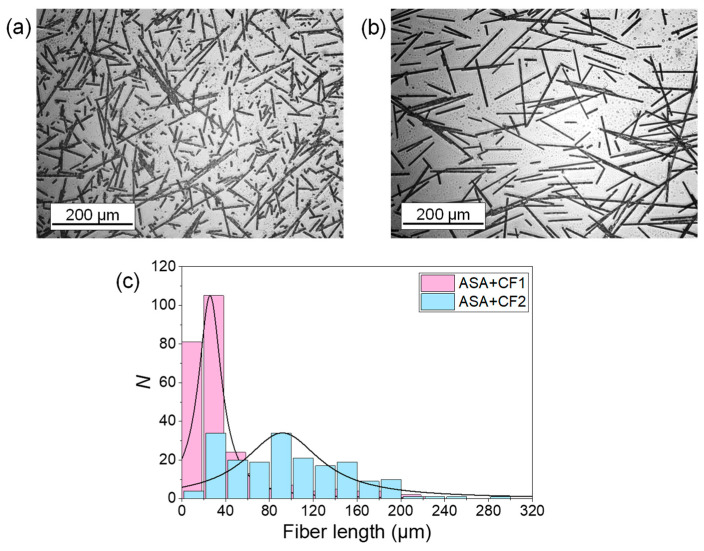
Optical microscopy images of the CF isolated from (**a**) ASA+CF1 and (**b**) ASA+CF2; (**c**) fiber length distribution of the CF of ASA+CF1 and ASA+CF2. In both cases *N* > 250. The CF length distribution was fitted to a Lorentz curve.

**Figure 2 polymers-16-00060-f002:**
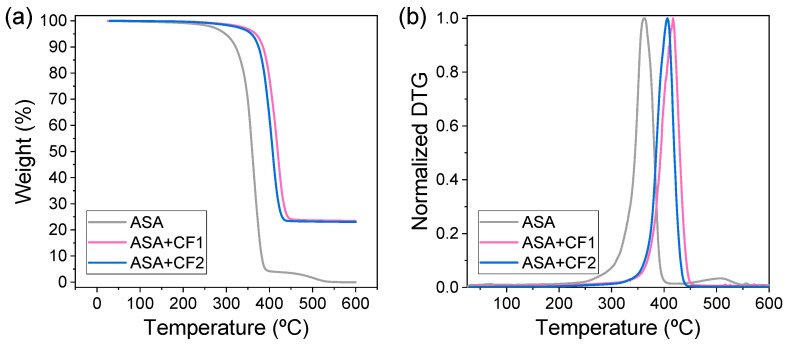
(**a**) TGA and (**b**) DTG curves of ASA, ASA+CF1, and ASA+CF2.

**Figure 3 polymers-16-00060-f003:**
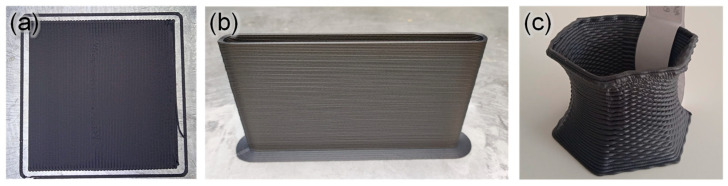
Digital photographs of (**a**) horizontal panel (XY plane); (**b**) vertical panel (XZ plane); and (**c**) a display vase (80 × 80 × 60 mm^3^) with different surface textures printed by FGF using ASA+CF1.

**Figure 4 polymers-16-00060-f004:**
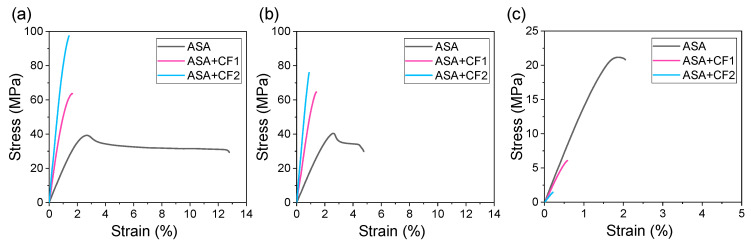
Representative stress–strain curves of ASA, ASA+CF1, and ASA+CF2 specimens prepared via (**a**) injection molding; (**b**) FGF (specimens oriented on the XY plane), and (**c**) FGF (specimens oriented on the XZ plane).

**Figure 5 polymers-16-00060-f005:**
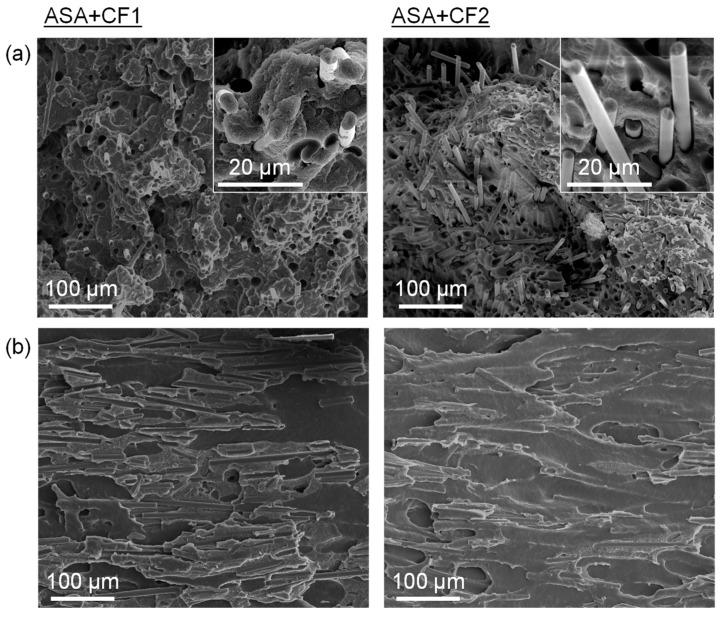
SEM micrographs of the (**a**) XY fracture surface and (**b**) XZ fracture surface of ASA+CF1 and ASA+CF2 tensile testing specimens manufactured via FGF. Insets in (**a**) illustrate the detail of the CF pull-out.

**Figure 6 polymers-16-00060-f006:**
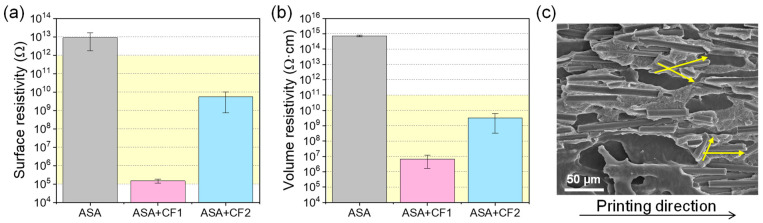
(**a**) Electrical surface and (**b**) volume resistivity of ASA, ASA+CF1, and ASA+CF2 standard specimens printed using FGF. The range of electrical resistivity values suitable for ESD materials is highlighted in light yellow in both cases. (**c**) SEM image of ASA+CF1. Yellow arrows show the overlapping of different CF, as a consequence of their deviation from the printing direction.

**Figure 7 polymers-16-00060-f007:**
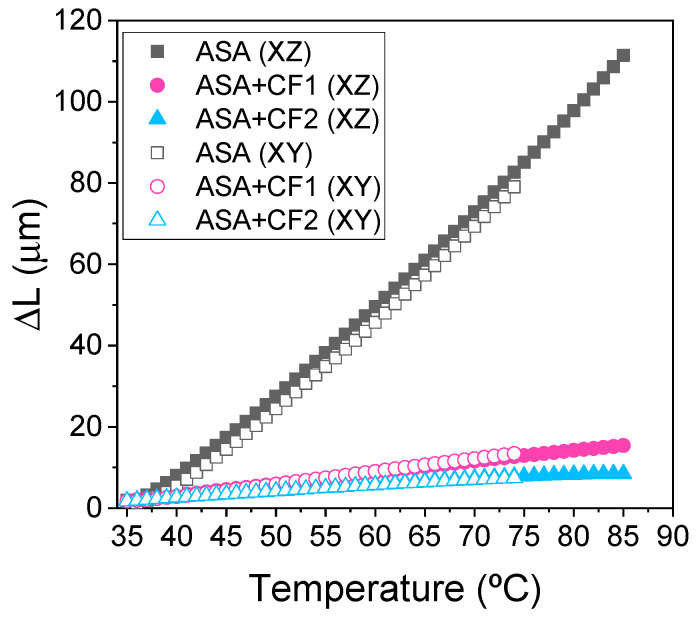
TMA curves of samples manufactured via FGF in the XY plane (FGF_XY) and XZ plane (FGF_XZ).

**Table 1 polymers-16-00060-t001:** MFR of the different materials used in FGF, measured at 245 °C.

	MFR (g/10 min)
ASA	44.3 ± 2.0
ASA+CF1	20.5 ± 1.9
ASA+CF2	10.3 ± 0.4

**Table 2 polymers-16-00060-t002:** Young’s modulus values from tensile testing curves of specimens manufactured via injection molding (IM) and FGF on the XY plane (FGF_XY) and XZ plane (FGF_XZ).

	Young’s Modulus (MPa)		
	IM	FGF_XY	FGF_XZ
ASA	2070 ± 40	1960 ± 120	1500 ± 50
ASA+CF1	5800 ± 400	5670 ± 430	1270 ± 60
ASA+CF2	8600 ± 380	9450 ± 400	700 ± 80

**Table 3 polymers-16-00060-t003:** Tensile strength values from tensile testing curves of specimens manufactured via injection molding (IM) and FGF on the XY plane (FGF_XY) and XZ plane (FGF_XZ).

	Tensile Strength (MPa)		
	IM	FGF_XY	FGF_XZ
ASA	39.3 ± 0.1	39.9 ± 0.6	21.2 ± 0.7
ASA+CF1	64.5 ± 2.7	65.4 ± 2.7	6.5 ± 1.1
ASA+CF2	95.2 ± 4.1	74.8 ± 4.3	1.2 ± 0.3

**Table 4 polymers-16-00060-t004:** Elongation at break values from tensile testing curves of specimens manufactured via injection molding (IM) and FGF on the XY plane (FGF_XY) and XZ plane (FGF_XZ).

	Elongation at Break (%)		
	IM	FGF_XY	FGF_XZ
ASA	13.5 ± 4.1	4.6 ± 0.7	2.1 ± 0.3
ASA+CF1	1.7 ± 0.1	1.4 ± 0.1	0.59 ± 0.08
ASA+CF2	1.4 ± 0.1	0.9 ± 0.1	0.21 ± 0.03

**Table 5 polymers-16-00060-t005:** CTE measured at 70 °C of ASA, ASA+CF1, and ASA+CF2 samples manufactured via FGF in the XY plane (FGF_XY) and XZ plane (FGF_XZ).

	CTE (µm/m·°C)	
	FGF_XY	FGF_XZ
ASA	75.5 ± 0.7	70.3 ± 0.6
ASA+CF1	12.1 ± 0.2	13.5± 0.9
ASA+CF2	7.4 ± 0.1	7.0 ± 0.3

## Data Availability

Data is contained within the article. Raw data will be made available by the authors upon reasonable request.

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
