# Peer review of "Influence of the Carbon Fiber Length Distribution in Polymer Matrix Composites for Large Format Additive Manufacturing via Fused Granular Fabrication"

_polymers, 2023, doi:10.3390/polym16010060_

Round 1

Reviewer 1 Report

Comments and Suggestions for Authors

The article titled “Influence of the Carbon Fiber Length in Polymer Matrix Composites for Large Format Additive Manufacturing by Fused Granular Fabrication” presents a very interesting research work. The experimental campaign is well-defined, the results are presented clearly and discussed well. There are little to no language errors in the article. However, the title and the scientific content of the article is misleading. The title makes the readers believe that the experimental results are reported for composites with CF of different length reinforced in them. But the distribution of the length of CF is quite wide to accept this title. I strongly recommend the authors to change the title from “Influence of the Carbon Fiber Length in…” to “Influence of the Carbon Fiber Length Distribution in…”.

Furthermore, there are some other issues that must be addressed. Therefore, I suggest a major revision before accepting the article for publication.

In Line 14, please replace “composite materials” with “composites”.

“Different authors proved that the percolation limit at… … is achieved at lower fiber concentrations when longer CF are used”. Please mention an average or median lower concentration reported in the literature.

How the fibre concentration of 23% was selected?

The number of composites with varying lengths tested is very limited. This is why this article cannot be considered as the one which studies the influence of CF length. So, redefine these statements throughout the discussion sections as CF length distribution.

“A display vase with different surface textures was also printed as a prototype using the “vase mode” unique contour”. This statement and the vase seem to be mispositioned in the article. Please consider the significance of this statement and decide whether it is necessary to be included or not.

“The melt flow rate (MFR) values of ASA, ASA+CF1 and ASA+CF2 were obtained in a LR-A001-A”. The experimental procedure and set up is not clear. Please explain it in detail.

“As expected, an increase in the thermal degradation of the composites when compared to pure ASA is observed”. The increase in thermal degradation temperature cannot be viewed clearly from the TGA curve. The temperatures are not very clear in Figure 2 either. Please provide a DTG curve for the thermogravimetric data to explain the degradation temperature.

Figure 1 clearly shows that there is a wide distribution of length in the CF fibres. Please consider this while revising the article.

What is the reason for cutting the tensile, electrical, and other test specimens from a printed block? Why couldn’t they have been printed directly for the required dimensions?

“On the contrary, the fibers are preferably displaced in an orientation perpendicular to the external force applied, not contributing to enhance the fracture mechanism”. This needs extensive explanation and supporting references.

Young’s modulus for the composite specimens oriented in XZ plane cannot be accepted. The stress-strain curve does not have enough strain value to calculate the Young’s modulus. Please check the standard used and consider revising this information.

In Figure 5a, fibre pull out cannot be observed.

“Both composites present typical values of semiconductors, indicating that the CF have virtually reached the percolation threshold, where a polymer matrix composite is no longer non-conductive”. Please support this statement with suitable references. In addition, the light yellow cannot be seen in Figure 6a.

Paragraphs in Lines 172-179 are repeated in Lines 189-196.

Please revise the whole discussion section while considering this study as the influence of CF length distribution. 

Author Response

The article titled “Influence of the Carbon Fiber Length in Polymer Matrix Composites for Large Format Additive Manufacturing by Fused Granular Fabrication” presents a very interesting research work. The experimental campaign is well-defined, the results are presented clearly and discussed well. There are little to no language errors in the article. However, the title and the scientific content of the article is misleading. The title makes the readers believe that the experimental results are reported for composites with CF of different length reinforced in them. But the distribution of the length of CF is quite wide to accept this title. I strongly recommend the authors to change the title from “Influence of the Carbon Fiber Length in…” to “Influence of the Carbon Fiber Length Distribution in…”.

We agree with the reviewer that this suggestion makes our manuscript more accurate, and we have modified this accordingly throughout the manuscript.

Furthermore, there are some other issues that must be addressed. Therefore, I suggest a major revision before accepting the article for publication.

In Line 14, please replace “composite materials” with “composites”.

Done.

“Different authors proved that the percolation limit at… … is achieved at lower fiber concentrations when longer CF are used”. Please mention an average or median lower concentration reported in the literature.

We have included the values reported in the different papers found in the literature. Since the values reported in the literature are so different, probably due to the strong influence of the CF length distribution, we have decided to give the full range of values found. We have also modified the discussion around this in the Results and discussion section.

How the fibre concentration of 23% was selected?

We did not optimized the CF content because the composites were directly supplied to us by the company. They distribute this material stating that this CF concentration is optimized to achieve an optimal performance of the material in additive manufacturing and injection molding processes.

The number of composites with varying lengths tested is very limited. This is why this article cannot be considered as the one which studies the influence of CF length. So, redefine these statements throughout the discussion sections as CF length distribution.

We have modified this throughout the manuscript.

“A display vase with different surface textures was also printed as a prototype using the “vase mode” unique contour”. This statement and the vase seem to be mispositioned in the article. Please consider the significance of this statement and decide whether it is necessary to be included or not.

The sentence has been rewritten and simplified.

“The melt flow rate (MFR) values of ASA, ASA+CF1 and ASA+CF2 were obtained in a LR-A001-A”. The experimental procedure and set up is not clear. Please explain it in detail.

We apologize for this. We have explained this in more detail.

“As expected, an increase in the thermal degradation of the composites when compared to pure ASA is observed”. The increase in thermal degradation temperature cannot be viewed clearly from the TGA curve. The temperatures are not very clear in Figure 2 either. Please provide a DTG curve for the thermogravimetric data to explain the degradation temperature.

We have included the DTG curve to better illustrate the increase of the thermal stability of the composites.

Figure 1 clearly shows that there is a wide distribution of length in the CF fibres. Please consider this while revising the article.

We acknowledge that there is a wide distribution of CF length in both cases. However, this something relatively normal after the processing a twin-screw extruder, since CF break into smaller pieces due to the shear forces that happen during the compounding. This must not be confused with the initial CF length, which is something that can be more controlled but it is beyond the scope of this paper since the composites were already provided by the company. Although this distribution is wide, it is also in the range of those reported by others (see refs. 23, 26 and 28 in the revised version of the manuscript).

What is the reason for cutting the tensile, electrical, and other test specimens from a printed block? Why couldn’t they have been printed directly for the required dimensions?

We do this to avoid mechanical defects associated with the printing process itself. This is a generally admitted procedure that has already been used in LFAM by others and us:

https://doi.org/10.1016/j.addma.2020.101255

https://doi.org/10.1016/j.compositesb.2023.110617

https://doi.org/10.1016/j.matdes.2020.108577

https://doi.org/10.1016/j.addma.2023.103908

“On the contrary, the fibers are preferably displaced in an orientation perpendicular to the external force applied, not contributing to enhance the fracture mechanism”. This needs extensive explanation and supporting references.

This has been rewritten. We have strengthened the discussion and have cited previous works published by us and others, where a similar effect was observed.

Young’s modulus for the composite specimens oriented in XZ plane cannot be accepted. The stress-strain curve does not have enough strain value to calculate the Young’s modulus. Please check the standard used and consider revising this information.

Young's modulus was measured according to the ASTM D638 standard, as the slope at 0.05%-0.25% strain. This was done in all cases except for ASA+CF2 XZ, which was measured up to its elongation at break value. We consider that this Young’s modulus is also valid for comparative purposes since the stress-strain curve is completely straight. In any case, we have added one sentence to clarify this.

In Figure 5a, fibre pull out cannot be observed.

Higher magnification images have been included in Figure 5a) as insets.

“Both composites present typical values of semiconductors, indicating that the CF have virtually reached the percolation threshold, where a polymer matrix composite is no longer non-conductive”. Please support this statement with suitable references. In addition, the light yellow cannot be seen in Figure 6a.

We have modified Figure 6 and rewritten this sentence, including some references.

Paragraphs in Lines 172-179 are repeated in Lines 189-196.

We apologize for this. We have deleted the duplicated paragraph.

Please revise the whole discussion section while considering this study as the influence of CF length distribution. 

The manuscript has been rewritten according to the reviewer’s suggestion.

Reviewer 2 Report

Comments and Suggestions for Authors

Manuscript Title: Influence of the Carbon Fiber Length in Polymer Matrix Composites for Large Format Additive Manufacturing by Fused Granular Fabrication

Authors: Pedro Burgos Pintos, Daniel Moreno Sánchez, Francisco J. Delgado, Alberto Sanz de León, Sergio I. Molina

Journal: Polymers (MDPI)

Manuscript id: polymers-2762465

Overall Comments

The authors have developed carbon fiber reinforced acrylonitrile-styrene-acrylate composites using larger format additive manufacturing based on Fused Granular Fabrication technique. The detailed influence of the size of the carbon fiber was studied to conclude that it has substantial influence on Young’s modulus, tensile strength, interlayer adhesion, coefficient of thermal expansion, and electrical properties. Finally, it was concluded that the length of the carbon fiber can be easily modified to tune the desired properties of the composite. The specific questions that the authors need to answer before publication are provided in the following part.

Specific questions

1.     The abbreviation of LFAM has been used in the Abstract, Line 15, without explaining the full form previously. The authors are requested to ensure all short forms or abbreviations are explained before their use in the manuscript.

2.     The lengths of the CFs in ASA-CF1 and ASA-CF2 need to be explained in Section 2.1, or if determined through tests, the same should be informed.

3.     Line 189 to 196 has repetitive statements, exactly similar to the previous paragraph before the figure. The authors are requested to proofread thoroughly.

4.     Lines 199 to 200 has repetition in the word “decrease”. Please rectify.

5.     Line 201 states, “In any case, all these values are in the range of the processability of materials for FFF and FGF”. What is this range? The authors are requested to provide this range with reference in the manuscript.

6.     The terms “this evidences” and “evidencing” do not seem fit. The authors are requested to use “this proves” or “proving” and similar conventional terms.

7.     Line 373 states, “…for ASA+CF1 and ASA+CF2 are below 15 and 8 μm/m·ºC for ASA+CF1 and ASA+CF2…”. Please remove duplicates.

8.     The authors have failed to explain the variation in CTE values obtained due to change in temperature. The authors are requested to explain the same thoroughly with proper justifications.

9.     Line 406 states, “that their length…”. It should be “than their length”.

The manuscript looks good and well written. However, the previous comments should be addressed before  can be accepted for publication.

Comments on the Quality of English Language

The English is good, but some changes are required. The authors are requested to proofread thoroughly before re-submission.

Author Response

Overall Comments

The authors have developed carbon fiber reinforced acrylonitrile-styrene-acrylate composites using larger format additive manufacturing based on Fused Granular Fabrication technique. The detailed influence of the size of the carbon fiber was studied to conclude that it has substantial influence on Young’s modulus, tensile strength, interlayer adhesion, coefficient of thermal expansion, and electrical properties. Finally, it was concluded that the length of the carbon fiber can be easily modified to tune the desired properties of the composite. The specific questions that the authors need to answer before publication are provided in the following part.

Specific questions

  1. The abbreviation of LFAM has been used in the Abstract, Line 15, without explaining the full form previously. The authors are requested to ensure all short forms or abbreviations are explained before their use in the manuscript.

We have included the definition of LFAM in the abstract.

  1. The lengths of the CFs in ASA-CF1 and ASA-CF2 need to be explained in Section 2.1, or if determined through tests, the same should be informed.

We have rewritten this, indicating how the CF length was measured in section 2.3.

  1. Line 189 to 196 has repetitive statements, exactly similar to the previous paragraph before the figure. The authors are requested to proofread thoroughly.

We apologize for this. The duplicated paragraph has been deleted.

  1. Lines 199 to 200 has repetition in the word “decrease”. Please rectify.

The sentence has been rewritten accordingly.

  1. Line 201 states, “In any case, all these values are in the range of the processability of materials for FFF and FGF”. What is this range? The authors are requested to provide this range with reference in the manuscript.

We have rewritten this, indicating that a threshold value of 10 g/10 min is generally admitted as the minimum value for processing materials by FFF or FGF.

  1. The terms “this evidences” and “evidencing” do not seem fit. The authors are requested to use “this proves” or “proving” and similar conventional terms.

We have modified these words throughout the manuscript, as suggested.

  1. Line 373 states, “…for ASA+CF1 and ASA+CF2 are below 15 and 8 μm/m·ºC for ASA+CF1 and ASA+CF2…”. Please remove duplicates.

The sentence has been rewritten deleting the duplicate content.

  1. The authors have failed to explain the variation in CTE values obtained due to change in temperature. The authors are requested to explain the same thoroughly with proper justifications.

We apologize for this. Our aim here is to compare the CTE of the different materials studied at a given temperature, proving that the presence of CF decreases the CTE to values similar to those of metals, making the ASA+CF composites an interesting alternative for low-temperature mold applications. To make our point clearer and simpler, we have deleted the values measured at 40 ºC, stablishing the CTE discussion around the values measured at 70 ºC.

  1. Line 406 states, “that their length…”. It should be “than their length”.

Done.

The manuscript looks good and well written. However, the previous comments should be addressed before  can be accepted for publication.

Reviewer 3 Report

Comments and Suggestions for Authors

The research article 'Influence of the Carbon Fiber Length in Polymer Matrix 2 Composites for Large Format Additive Manufacturing by Fused Granular Fabrication' aims to evaluate the influence of the fiber length in the mechanical and functional properties of printed parts.

2.1. Materials

The materials section is confusing. The authors refer to ASA and ASA loaded with 23% by weight of carbon fiber (CF) composite granules that were supplied by the industry. But, 'Two compounds with different CF lengths...' CF lengths? What CF lengths were obtained? Clarify..

L101 'ASA-CF1 and ASA-CF2 were provided...' This line is very confusing. Authors should clarify the sentence and adequately explain the materials used.

   CF1 and CF2?? Clarify what you are trying to express.

L121. a Babyplast 10/12P machine (Cronoplast SL, Spain)… the authors should clarify this point. Babyplast is from Italy.

3. Results and discussion

Figure 4 should be represented using the same scale as the manuscript attempts to compare the mechanical effects of the manufacturing and orientation processes.

The authors must explain why Young's modulus of FGF_XY is higher than that obtained by injection.

If the diameter of the CF is 5 ± 1 µm, and the average lengths of the CF are 42 ± 41 µm and 96 ± 57 µm. How can the authors suggest or claim the representability or repeatability of the results with such a large particle dispersion?

L164-170 the authors assume a lot about the materials' provenance and composition. It is not clear whether they are vague assumptions or they know clearly. This situation prevents the repeatability of the experiment and digresses about the results. What implications would these arguments have on the thermal properties? Is there more degradation in CF1 than in CF2 or vice versa?

The fabrication of the pellet composites probably has much involvement in the structure and materials properties, so the authors must write the manuscript more truthfully.

Author Response

The research article 'Influence of the Carbon Fiber Length in Polymer Matrix 2 Composites for Large Format Additive Manufacturing by Fused Granular Fabrication' aims to evaluate the influence of the fiber length in the mechanical and functional properties of printed parts.

2.1. Materials

The materials section is confusing. The authors refer to ASA and ASA loaded with 23% by weight of carbon fiber (CF) composite granules that were supplied by the industry. But, 'Two compounds with different CF lengths...' CF lengths? What CF lengths were obtained? Clarify..

L101 'ASA-CF1 and ASA-CF2 were provided...' This line is very confusing. Authors should clarify the sentence and adequately explain the materials used.

   CF1 and CF2?? Clarify what you are trying to express.

In this section we just want to indicate that the two different composites were provided by the company. The only information given by the company is that both have the same carbon fiber content, but different average fiber length. To quantify the fiber length distribution, we analyzed the commercial pellets by microscopy, as indicated in section 2.3. We have rewritten this part trying to clarify this.

L121. a Babyplast 10/12P machine (Cronoplast SL, Spain)… the authors should clarify this point. Babyplast is from Italy.

We have revised and modified this accordingly.

  1. Results and discussion

Figure 4 should be represented using the same scale as the manuscript attempts to compare the mechanical effects of the manufacturing and orientation processes.

We agree with the reviewer that, in principle, the three graphs included in Figure 4 should be represented with the same scale. However, when this is done in the case of Figure 4c), the curves are no longer clearly visible. Therefore, we have decided to show Figures 4a) and 4b) with the same scale but to leave Figure 4c) as is. We believe this way the information is clearer. Moreover, Tables 2-4 also allow to compare appropriately the mechanical properties of the materials.

The authors must explain why Young's modulus of FGF_XY is higher than that obtained by injection.

There is a preferential direction of alignment of the CF in the FGF process, while in injection molding they are oriented more randomly. We believe that this may contribute to increasing the stiffness of the material. We have included a sentence in this regard and cited a paper where authors observed a similar behavior.

If the diameter of the CF is 5 ± 1 µm, and the average lengths of the CF are 42 ± 41 µm and 96 ± 57 µm. How can the authors suggest or claim the representability or repeatability of the results with such a large particle dispersion?

We acknowledge that there is a wide distribution of CF length in both cases, but we consider that they are different enough to stablish differences in terms of mechanical and functional properties.

These wide distributions are something relatively normal after the processing in a twin-screw extruder, since the CF break into smaller pieces due to the shear forces that happen during the compounding. In fact, the CF length distributions obtained are comparable to those previously reported by others (see refs. 23, 26 and 28). Moreover, following Reviewer1’s advice, we now discuss the CF length distribution rather than the CF length.

L164-170 the authors assume a lot about the materials' provenance and composition. It is not clear whether they are vague assumptions or they know clearly. This situation prevents the repeatability of the experiment and digresses about the results. What implications would these arguments have on the thermal properties? Is there more degradation in CF1 than in CF2 or vice versa?

The fabrication of the pellet composites probably has much involvement in the structure and materials properties, so the authors must write the manuscript more truthfully.

The details about the manufacturing of the materials are confidential information on the part of the company. Since we cannot state anything about the reasons why the composites supplied have different fiber lengths, we have decided to remove this in the revised version so as not to confuse the reader. We would like to point out that our aim with this paper is to correlate the structure and materials properties with the processing by large format additive manufacturing, not with the fabrication of the composite. The reproducibility of the results does not have to be a problem since all the materials used in this study can be purchased.

The degradation of the ASA+CF1 and ASA+CF2 composites is discussed from the TGA results. We have observed a clear increase in the maximum degradation temperature for both composites with respect to ASA, but we did not observe significant differences between ASA+CF1 and ASA+CF2.

Round 2

Reviewer 3 Report

Comments and Suggestions for Authors

The article Influence of the Carbon Fiber Length Distribution in Polymer Matrix Composites for Large Format Additive Manufacturing  by Fused Granular Fabrication is a reviewed manuscript that focuses on the  influence of the fiber length distribution on the mechanical and functional properties of printed parts.

The authors have revised the manuscript properly for publication